# Lessons Learnt from Engineering Science Projects Participating in the Horizon 2020 Open Research Data Pilot

**Timothy Austin** [1,*,†,‡]**, Kyriaki Bei** [2,†]**, Theodoros Efthymiadis** [2,†] **and Elias P. Koumoulos** [2,*,†]

¹  Joint Research Centre (JRC), European Commission, 1755 LE Petten, The Netherlands
²  Data Science Group, Innovation in Research & Engineering Solutions (IRES), Rue Koningin Astridlaan 59B, 1780 Wemmel, Belgium; sbei@innovation-res.eu (K.B.); tefthymiadis@innovation-res.eu (T.E.)
*  Correspondence: simon.austin@ec.europa.eu (T.A.); epk@innovation-res.eu (E.P.K.)
†  These authors contributed equally to this work.
‡  The views expressed are purely those of the author and may not in any circumstances be regarded as stating an official position of the European Commission.

**Abstract:** Trends in the sciences are indicative of data management becoming established as a feature of the mainstream research process. In this context, the European Commission introduced an Open Research Data pilot at the start of the Horizon 2020 research programme. This initiative followed the success of the Open Access pilot implemented in the prior (FP7) research programme, which thereafter became an integral component of Horizon 2020. While the Open Access phenomenon can reasonably be argued to be one of many instances of web technologies disrupting established business models (namely publication practices and workflows established over several centuries in the case of Open Access), initiatives designed to promote research data management have no established foundation on which to build. For Open Data to become a reality and, more importantly, to contribute to the scientific process, data management best practices and workflows are required. Furthermore, with the scientific community having operated to good effect in the absence of data management, there is a need to demonstrate the merits of data management. This circumstance is complicated by the lack of the necessary ICT infrastructures, especially interoperability standards, required to facilitate the seamless transfer, aggregation and analysis of research data. Any activity aiming to promote Open Data thus needs to overcome a number of cultural and technological challenges. It is in this context that this paper examines the data management activities and outcomes of a number of projects participating in the Horizon 2020 Open Research Data pilot. The result has been to identify a number of commonly encountered benefits and issues; to assess the utilisation of data management plans; and through the close examination of specific cases, to gain insights into obstacles to data management and potential solutions. Although primarily anecdotal and difficult to quantify, the experiences reported in this paper tend to favour developing data management best practices rather than doggedly pursue the Open Data mantra. While Open Data may prove valuable in certain circumstances, there is good reason to claim that managed access to scientific data of high inherent intellectual and financial value will prove more effective in driving knowledge discovery and innovation.

**Keywords:** Horizon 2020; data management plan; advanced characterisation; interoperability; materials properties; digitisation

## 1. Introduction

The engineering materials sector has relevance for all industrial domains, from electronics and nanotechnologies to energy production and aerospace. This circumstance is reflected in the research Framework Programmes of the European Union whereby very significant resources are invested in numerous multi-partner projects that aim to develop and qualify novel and advanced materials. Such projects tend to generate significant

volumes of data that are of high inherent intellectual, financial and potentially commercial value. Consequently, any initiative aimed at improved data management has the potential to add value to such projects, both during and beyond the term of the project.

At the European Commission, Neelie Kroes was instrumental in voicing the importance of data in driving the digital economy [1], and subsequent data policies were intended to ensure that European industrial and scientific entities take advantage accordingly [2–4]. Most recently, the European industrial and data strategies [5,6] recognise the fundamental role of data in a society that is entirely reliant on digital systems, whereby the latter emphasises the need for data standards and anticipates the creation of ten sectorial data spaces. When giving consideration to the topic of digital transformation in her first state of the union address, Ursula von der Leyen gave attention to this policy imperative by highlighting that "industrial data is worth its weight in gold when it comes to developing new products and services. But the reality is that 80% of industrial data is still collected and never used. This is pure waste." [7]. The reasons for this circumstance are debatable but certainly the lack of a joined-up infrastructure for industrial data is an obstacle to data reuse.

Although a welcome development, the recognition of the worth of data follows decades of neglect and hence concrete measures to promote responsible and effective practices are only recently beginning to emerge, such as the introduction of data management policies by funding agencies [8] and publishing houses [9,10]. Even so, very significant challenges remain if effective data management is to become a feature of mainstream industrial and scientific processes. To a greater extent, the FAIR data principles [11] offer a solution to this challenge, whereby findability, accessibility, interoperability and reusability are identified as the core 'attributes which are essential to extract the full scientific value from data resources and to unleash the potential for large-scale, machine-driven analysis' [12]. In this context, the Horizon 2020 Open Research Data pilot was not simply an exercise to promote responsible management of the research data but was an initiative with immediate relevance to policy imperatives in the industrial and scientific sectors.

Following the success of the Open Access pilot implemented in Framework Programme 7 (FP7), the European Commission introduced the Open Research Data (ORD) pilot at the start of the Horizon 2020 (H2020). This is a welcome development given that research data are at the foundation of the scientific process, providing the basis for analysis, reasoning and discussion. Examples of research data include statistics; the results of experiments; measurements; observations resulting from fieldwork; survey results; interview recordings and images.

The H2020 ORD pilot is just one of many examples of funding bodies mandating data management as a prerequisite for funding. Many publication houses also place similar demands on prospective authors. Initially, the H2020 ORD pilot was limited to certain calls for proposals, with the option for any project to participate voluntarily. Thereafter, from 2017 the ORD pilot became a feature of all calls. With the terms of participation in the ORD pilot defined in Article 29.3 of the Model Grant Agreement (MGA), the delivery at month six of a data management plan (DMP) is a mandatory requirement. Although delivered at an early stage, the DMP is expected to evolve over the term of the project.

Unless deciding to opt out for very specific reasons, all H2020 projects are participating in the ORD pilot, which aims to improve and maximise the access to and reuse of research data. Issues on openness, protection of scientific information and other privacy concerns are also addressed by the ORD pilot, which advocates adhering to the principle of data being "as open as possible, as closed as necessary" [13]. The ORD pilot applies primarily to the data needed to demonstrate the credibility of research and specifically to the data required to validate results reported in scientific publications. In this circumstance, Open Access to the minimum necessary research data is often sufficient, whereas more restrictive access can be employed to avoid the disclosure of data subject to different considerations, such as intellectual property rights (IPR) or further

exploitation, noting that irrespective of an open or closed access policy, the data can be protected or unprotected. Thus, the European Commission takes a pragmatic approach towards access and recognises there are legitimate reasons for some or even all research data generated in a project to be kept closed or embargoed. Hence, while one goal of the ORD pilot is to make more data publicly available, the data access needs to serve the interests of all stakeholders. In other words, if more value or a different kind of value can be derived from the data remaining closed, it is reasonable for data access to be restricted. Similarly, if there are risks associated with the open availability of data, it is again reasonable for data access to be restricted. Both these circumstances have been encountered in the projects in which the authors have participated. As will become apparent, the issue of access is a little more nuanced than simply open or closed and instead there are convincing arguments in favour of managed data access.

The ORD requirement for delivery of a DMP during the early stages of a project provides a clear indication of its important role in promoting effective data management. The DMP describes the life cycle of the data to be generated, collected and processed by an H2020 project. The DMP is also important for the credibility of the data produced throughout a project, whereby it offers the European Commission, its expert evaluators and the public (in the circumstance the DMP is openly available) an overview of the progress and the consistency of the research. As a result, the European Commission suggests that a DMP is submitted even if a project has opted out of the ORD pilot [13].

In summary, the ORD pilot is intended to make scientific data more widely available. To have value and become useful to others, such data needs to meet specific requirements in accordance with the FAIR principles. To this end, a DMP allows all stages of the data lifecycle to be documented so that a permanent record exists of the data that are generated, processed or collected throughout the project. Furthermore, data handling procedures during and beyond the term of the project can be described, whereby information on methodologies and standards applied to the data are an important consideration. Finally, the DMP needs to give attention to the dissemination level and whether data access is open or restricted, preferably with justifications.

## 2. Framework and Practice

### 2.1. FAIR Data in Practice

The FAIR data principles respond to a need to improve and maximise access to research data and improve the infrastructure supporting its reuse. These principles are the result of collaboration between academia, industry, funding agencies and scholarly publishers and are intended as a guideline for project partners who wish to share their data and make them available for reuse by other beneficiaries [11,12,14]. The principles of FAIR data aim to create data that are findable, accessible, interoperable and reusable by people and machines [15–17], as follows:

- Findable—data and supplementary materials have sufficiently rich metadata and a unique and persistent identifier;
- Accessible—metadata and data are understandable to humans and machines. Data are deposited in a trusted repository;
- Interoperable—metadata use a formal, accessible, shared and broadly applicable language for knowledge representation;
- Reusable—data and collections have a clear usage license and provide accurate information on provenance.

To achieve these objectives, the DMP of any project should thus include all necessary actions (Table 1) towards creating FAIR data.

Table 1. Actions needed for meeting FAIR requirements.

| Findable | Accessible | Interoperable | Re-usable |
|---|---|---|---|
| - Data and metadata are assigned a globally unique and eternally persistent identifier. | - Data and metadata are retrievable by their identifier using a standardised communications protocol. | - Data and metadata are retrievable by their identifier using a standardised communications protocol. | - (Meta)data use a formal, accessible, shared and broadly applicable language for knowledge representation |
| - Data are described with rich metadata. | - The protocol is open, free and universally implementable. | - The protocol is open, free and universally implementable. | - (Meta)data use vocabularies that follow FAIR principles |
| - Data and metadata are registered or indexed in a searchable resource. | - The protocol allows for an authentication and authorisation procedure, where necessary. | - The protocol allows for an authentication and authorisation procedure, where necessary. | - (Meta)data include qualified references to other (meta)data |
| - Metadata specify the data identifier. | - Metadata are accessible, even when the data are no longer available. | - Metadata are accessible, even when the data are no longer available. | - (Meta)data use a formal, accessible, shared and broadly applicable language for knowledge representation |

Despite the efforts of the scientific community to formulate all aspects of the FAIR principles, they remain open to interpretation. This is to be expected because they are intended as guidelines rather than a standard or technology [18]. Consequently, evaluating FAIRness is potentially controversial [19]. Even so, the perceived value of quantifying compliance with FAIR principles has resulted in a design framework and exemplar metrics for FAIRness [20] and a corresponding implementation [19,21].

In the context of promoting FAIRness, the experience of the authors is that data citation is particularly effective, whereby any data set, whether restricted or open, can be discovered. Thereafter, data reuse can be managed by way of licensing and accredited by way of entries in the references section of a traditional report or scientific publication. Data citation thus entirely addresses the findable and accessible strands of the FAIR principles and motivates data sharing (and hence reuse) by ensuring accreditation. As an example, the ODIN Portal at https://odin.jrc.ec.europa.eu (accessed 30 October 2020) serves the data management requirements of the referenced projects where the JRC participates in a data management capacity and hosts many thousands of citable data sets. Previous FAIR evaluations [19] have yielded a nominal score of about 50% for the citable data, noting that the score is considered nominal because the FAIR Maturity Evaluation Service overlooked certain characteristics of the ODIN data, including licensing and distributions compliant with emerging data standards. The results of two such evaluations dating to June 2020 (one for a citable data set and the other for a citable data catalog) are posted at https://fairsharing.github.io/FAIR-Evaluator-FrontEnd/#!/evaluations/4079 and https://fairsharing.github.io/FAIR-Evaluator-FrontEnd/#!/evaluations/4080, respectively. By implication, given that the ODIN data sets have a consistent structure and content, this measure of FAIRness can reasonably be claimed for any project or publication the data for which are hosted at the ODIN Portal.

While the maturity of frameworks such as DataCite [22] means that data citation can be implemented without undue difficulty and thereby address the findable and accessible components of the FAIR principles, some considerable effort is (and will be) required to

address the interoperable and reusable components of the FAIR principles [15,23]. For data to be interoperable in any given domain requires data standards that are (1) technically robust; (2) fit for purpose; (3) find widespread adoption; and (4) are maintained over the longer term, which in turn demands a long-term intellectual and financial investment.

As data become more accessible and interoperability standards ease data transfer and aggregation overheads, more opportunities for data reuse will arise. In this circumstance, due consideration of the criteria that will motivate researchers to reuse data is needed. Clearly, time and cost savings are a key motivator. However, no researcher, irrespective of the time or cost savings, will rely on data for which the quality and provenance cannot be demonstrated. Quality assurance thus becomes a key factor in determining potential for reuse. While there are many aspects of data creation that can contribute to a quality index, such as the extent of metadata and the availability of and adherence to documented procedures and protocols, the circumstance of peers having reviewed data will prove influential. Thus, data assessment by subject matter experts has a pivotal role to play in promoting reuse. However, given that the resources invested in improved data management will undoubtedly be at the expense of other research activities, it is unrealistic to expect that data review can be undertaken in the same manner as traditional peer-review (of scientific publications). Instead, data review will need to rely largely on automated procedures, e.g., checks on mandatory data and metadata and other novel solutions.

### 2.2. DMP in Practice—The Case of EC-Funded Projects

With a view to gaining insights into the impact of the ORD pilot, consideration has been given to the data management activities and outcomes of 10 engineering materials research projects where the authors have led the data management activities. All projects delivered (and in large part maintained) a DMP, some of which adhered strictly to the H2020 template, while others evolved into comprehensive records of the testing campaign. The European Commission provides a DMP template, which is regarded as a baseline practical guide for projects opted in the ORD pilot [13,24]. It is divided into six broad categories of meta-information on the project data. The template indicates the various considerations that deserve attention, as follows:

1. Data Summary
    a. Purpose of the data collection/generation and its relation to the objectives of the project;
    b. Types and formats of data generated/collected;
    c. Information on use of any existing data;
    d. Origin of the data;
    e. Expected size-volume of the data;
    f. Data utility—beneficiaries;
2. FAIR Data;
3. Allocation of Resources:
    a. Costs for making data FAIR;
    b. Responsible partner/individual for data management of the data of subject;
4. Data Security:
    a. Provisions in place for data security (including data recovery as well as secure storage and transfer of sensitive data);
    b. Information about safe storage, long-term preservation and curation;
5. Ethical Aspects:
    a. Ethical or legal issues that can have an impact on data sharing;
6. Other Issues.

With the scope of the H2020 DMP template extending to a data summary; FAIR data; allocation of resources; data security; and ethical aspects, it effectively defines the data management policy of the project and the implementation of that policy. Furthermore, if the

DMP is utilised to co-ordinate the testing campaign, its scope can extend to test matrices, data protocols, software development, etc. Depending on the data management needs and objectives of each project, the structure of its DMP can be tailored accordingly [25].

Although the management structure of any particular project does not need to include a body with oversight of the data management policy and its implementation, several projects chose to establish a Data Committee, the inspiration for which comes from the reference to a 'data access committee' in the H2020 DMP template [13] but with a broader remit. Thus, the tasks performed by a Data Committee can include maintaining the DMP; formulating data protocols in support of the research objectives of the project; developing data access and data sharing policies; and reviewing data. Where data review has been undertaken, it has typically relied on a degree of automation and has either been limited to a cursory examination to confirm that mandatory data have been made available or otherwise involved a somewhat time-consuming evaluation of data reports [24].

### 2.3. EC-Funded Engineering Materials Projects

The material science and engineering domain produces large volumes of data of significant inherent intellectual and financial worth, but its data management infrastructure is less developed than that of other domains, such as the humanities, life sciences and natural sciences. The material science and engineering domain thus stands to benefit from improved data management practices.

The primary interest of material scientists and engineers is to understand the relationships between material structure and performance with a view to identifying and developing materials best suited to a particular application. In this regard, both structure and performance are described by qualitative and quantitative properties. There are many experimental techniques, otherwise called characterisation and qualification methods. These methods have been developed to obtain high accuracy measurements of the various material properties under examination, including the evaluation of mechanical, physical, functional and electronic properties; surface characteristics; bulk structure; and composition. Moreover, the experimental procedures are complemented by simulations from the field of materials modelling. These simulations are used to estimate the values of certain material properties *a priori* and their implementation requires a physical model, a mathematical solver and a software to execute the calculations. Finally, the materials manufacturing domain involves various synthesis and forming processes that take place under certain physical conditions in order to produce materials with the desired properties.

Modelling experts collect requirements and specifications of the materials and products that will be manufactured and run simulations to provide insights and guidance to the manufacturing process that will follow. The output of the simulations is heavily reliant on the chosen physics model and assumptions, the mathematical solver and the software package. The input of the modelling team combined with the expertise of the manufacturing team allows samples or products to be manufactured and thereafter examined by characterisation experts to assess their performance. This allows the modelling and manufacturing activities to be optimised based on the measured properties of the manufactured objects.

The larger part of most materials research projects will be concerned with some or all the aforementioned activities and hence it is reasonable to claim that the larger part of the research investment is allocated to data creation. With H2020 engineering materials projects typically valued at between EUR 5 and EUR 10 million and involving upwards of 10 partners, there is clearly an opportunity for and a case in favour of establishing a well-developed data management culture.

Whereas the engineering sciences stand to benefit from improved data management practices, the accompanying challenges are significant. Consortia are comprised of partners from different disciplines using various physics models, mathematical solvers, software products, manufacturing technologies, on-line monitoring systems and characterisation instruments and protocols. Typically, equipment comes with its own

specialised data format that is not interoperable across systems. Moreover, different experts often understand and interpret material science concepts differently, something that might become evident when observing the diverse data documentation and knowledge representation approaches. To address these challenges, several H2020 projects in which the authors have participated, such as INCEFA-PLUS [26], M4F [27] and OYSTER [28], have given attention to developing interoperability standards and semantic technologies for data.

### 2.4. Author-Led ORD Activities

For those projects where IRES led the ORD tasks, the preliminary version of the DMP relied on a survey of project partners, whereby one questionnaire was completed for each new data set generated within the project. This allowed IRES to contact partners at an early stage of the project and gain an understanding of their data creation processes. Thereafter, having drafted a preliminary DMP, partners were invited to complete a DMP questionnaire for each one of their data creation processes that they were willing to share. The results of these surveys allowed IRES to provide partners with informed guidance on how to make their data FAIR. This continues until the data meet the FAIR requirements and can be published together with metadata information.

In parallel with the described DMP methodology, additional approaches were adopted according to the requirements of individual projects. In the SMARTFAN project [29] for example, an online platform was developed to store and share the data generated by the partners. Thereafter, experiences gained operating the SMARTFAN platform allowed more sophisticated data platforms to be developed for the DECOAT [30] and REPAIR3D [31] projects, whereby the conflicting requirements for efficient data documentation and limited additional effort for the partners were taken into consideration. The process of balancing the two needs resulted in a simplistic user interface to organise the data and assign the minimum necessary metadata. The main operation of the two platforms is depicted in a workflow form in Figure 1.

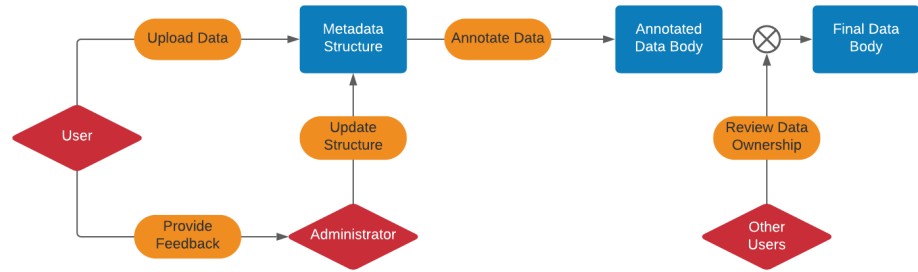

**Figure 1.** Data management platform workflow for the DECOAT and REPAIR3D projects.

The primary purpose of the platforms is to assign metadata to the data sets of the project partners. This is achieved using a predefined metadata structure developed by the platform administrator (IRES) using domain expertise. The partners use this structure to annotate their data. They are also encouraged to provide feedback on how to update the metadata structure throughout the project. After the completion of the data annotation process, project partners are requested to review the annotated data sets, especially those that are marked as public and intended to be shared after the project. In the case of ownership conflicts or violations, each partner is able to raise objections regarding the data access model of the data sets of interest. After taking such objections into consideration, the data provider and, when necessary, the platform administrator adjust the access model accordingly.

Additionally, in the REPAIR3D project an application ontology was developed to describe the complicated concepts of the project, which involves the development of a

number of products from plastic waste and the extensive study of their recycling cycles. Moreover, in the DECOAT project, additional effort was deemed necessary to pass from basic principles to the usefulness of such a tool, finally ensuring that the consortium would contribute to the best of their benefit. To address this, an educational workshop was given in the form of an online webinar in order to facilitate partner engagement in the data management process. This workshop provided useful insights on data management, data set identification and FAIR principles, especially the ones of findability and interoperability. Furthermore, in the LIGHTME Open Innovation Test Bed [32] project, an extensive mapping of the data sources of the different partners was conducted and a data harmonisation strategy, as well as data uploading procedures and a relational database schema for the Open Innovation Environment were proposed. Finally, in the OYSTER project [28], which is strongly focused on data standardisation, the DMP included the adoption of specialised domain-specific material characterisation data vocabularies. In that regard, the vocabularies were integrated in the questionnaires of the DMP and published in Zenodo [33] to serve as a standard for data management in materials characterisation. The data schema that was created through the combination of the DMP concepts and the characterisation data vocabulary was also encoded in the form of an ontology [34] that aims to harmonise the top-level concepts of data management with the domain specific concepts of the material characterisation domain.

For those projects where the JRC led the ORD tasks, as well as providing a means to define and implement a data management policy, the DMP often served as a container for all testing documentation, hence contributing to the co-ordination of the testing programme. For all projects, the DMP adhered strictly to the H2020 template, noting that the data summary and FAIR data sections occupied the larger part of the document. With a view to project partners being engaged from the outset, the project proposal included a near-complete draft of the DMP. In the circumstance of a favourable evaluation, this circumstance alleviated the work to submit the DMP deliverable at month six. Thereafter, the JRC typically recommended (1) to appoint a Data Committee with responsibility for various tasks and (2) to extend the purpose of the DMP to documenting the test campaign in its entirety. For this latter, annexes served the purpose of documenting data protocols, software development, certificates of conformance, specimen drawings, etc. The data access level was typically declared as restricted at the outset of the project, with a decision about Open Access scheduled for midterm [35].

## 3. Discussion

A DMP serves the objectives of the H2020 ORD pilot by promoting the FAIR data principles and fostering improved data management practices. In turn, making data FAIR renders them more easily available to other researchers, potentially resulting in the generation of new scientific knowledge. While the ever-increasing body of data will be of value to the scientific community, the individual researchers who produce the data may question the value of data management because the required time, effort and resources could be perceived as detracting from their research, whilst benefitting others. Any such misgivings are likely to be further compounded by the increased transparency and scrutiny that will result from easier access to research data. Consequently, attention needs to be given to the workflows, infrastructure and incentives needed to convince the research community of the value of data management.

### 3.1. Challenges

While there are convincing arguments that favour data management becoming a useful (and possibly indispensable) feature of the mainstream research process, the accompanying risks and pitfalls need to be acknowledged and accommodated.

*Research culture*—until recently, the research community has operated to good effect in the complete absence of any systematic approach to data management. That said, it is reasonable to assume that the failure of the scientific community to preserve (and enable

access to) the larger proportion of the considerable body of data it has generated has impeded scientific progress because the circumstance prevents the re-examination and reuse of data and risks duplication of effort. Even so, such arguments may still be insufficient to motivate project partners to engage in data management activities. There is thus a need to demonstrate to researchers the value of data management, such as data quality improvement; minimisation of data loss; better utilisation of resources; re-examination and reuse; and increased citations.

*Data sharing*—ultimately, scientific research aims to benefit society and while this principle applies equally to data sharing practices, it is not necessarily the case that Open Data always satisfies this requirement. Although the H2020 ORD principle of "as open as possible, as closed as necessary" encourages partners to make data open to third parties, partners may be reluctant to engage in the activities required by the DMP. For example, there is the risk that the term "open" can be (mis)interpreted as mandatory, which may act as an obstacle to organisations participating in the pilot.

Even where there is broad support for data management as a core component of a project, it only takes one dissenting partner to undermine the entire initiative because the resulting body of data is incomplete and hence of limited value. Moreover, there is the risk of data sharing being restricted to subgroups of partners, thereby leading to fragmented cooperation, potentially alienated from the rest of the consortium. This claim is supported by evidence from seven projects (of more than 50 unique partners) where IRES explored the data access preferences of the different project partners, asking them if their data sets should be open (public), restricted (shared within the consortium) or closed (not shared at all). The results are summarised in Figure 2.

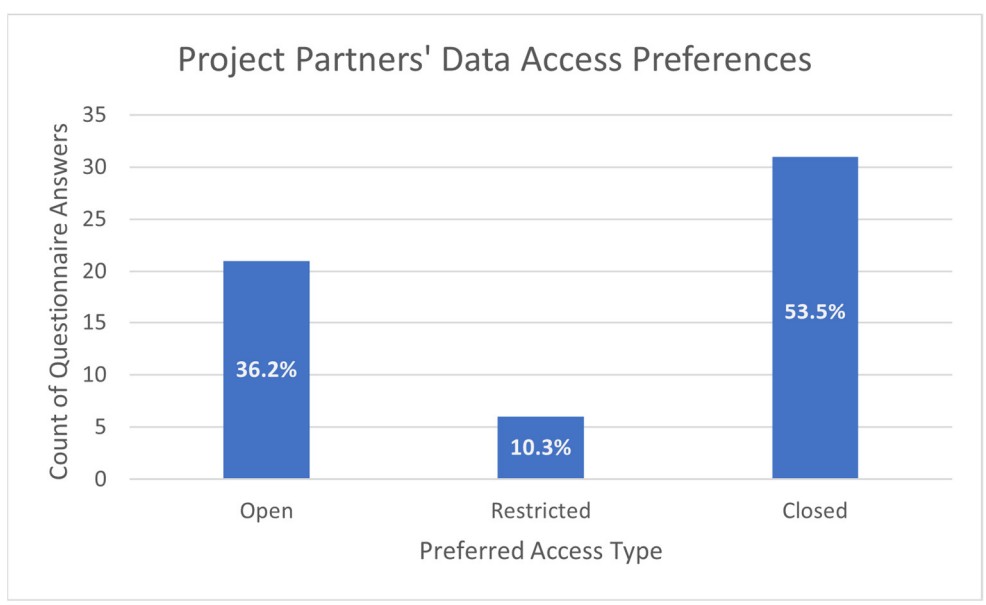

**Figure 2.** Data access model in IRES-led activities.

Despite being protected by the terms of the Grant Agreement and Consortium Agreement, the results indicate an unwillingness of more than half the project partners to share their data even within the project consortium. Instead, they prefer to avoid data sharing wherever possible unless it is essential to achieve certain project objectives. In those cases, they opt only to share their data with targeted partners to complete the project tasks. In turn, such circumstances risk undermining the co-ordination of the project.

Another potentially problematic feature of data sharing is the increased transparency and scrutiny. Scientific data are often complex and generated by emerging protocols, i.e., not following the recognised standards. By sharing their data, whether openly or to a lesser extent as part of a managed co-operation, researchers are exposing their data and

related findings to potential criticism. Furthermore, there is the potential for the misinterpretation of results. The authors have encountered such concerns, which have, on occasion, undermined the data management activity.

*Infrastructure*—another obstacle to data management is the likely absence of infrastructure, including interoperability standards, that is required. It is not unreasonable to claim that the pursuit of Open Data has been motivated by the success of the Open Access initiative and whereas the infrastructure and workflows for the latter have been established over many hundreds of years, data management infrastructures and workflows are only just emerging. Thus, while expectations of the benefits arising from improved data management are high, much work is required to establish a robust infrastructure on which to build a vibrant data management ecosystem. In turn, the time and effort needed to realise such an infrastructure risks losing momentum and enthusiasm. Furthermore, even with the establishment of a robust infrastructure, until the various scientific disciplines have developed the data technologies (i.e. interoperability standards) required for seamless data exchange and aggregation, data gathering tasks will remain a significant burden that detracts from rather than contributes to the research process.

*Project coordination*—perhaps of most immediate concern are the practical problems the authors have encountered relating to resourcing and coordination. When drafting a proposal, there is the risk that consortia pay insufficient attention to the DMP and data management. This circumstance is understandable given that investing resources in Open Access and data management could be perceived as detracting from the research. Unfortunately, the outcome is that authoring the DMP (and hence formulating the data management policy) is often left to a single partner. Thereafter, during the project implementation, other partners are unaware of their responsibilities. This circumstance can lead to problems that risk undermining the data management component of a project. Firstly, most of the partners remain unaware of the expected data flow during the project and their role or responsibility in that process, which leads to misunderstandings about the handling of the data during the active phase of the project. Second, the resources allocated to DMP tasks are often insufficient to account for the person months and infrastructure needed for their implementation. As a result, partners may find themselves lacking the necessary resources to cover the data management activities throughout the term of the project. The authoring of the DMP by a single partner also risks misunderstanding about the data to be shared between project partners, as well as the expected data quality. In addition, if data are not handled in a standardised way, sharing among the partners will be difficult, resulting in ineffective use of resources and delay the outcome of the research.

The DMP may itself also present obstacles. For example, academic and industrial partners alike might have good reason to keep their data closed and therefore be resistant to participation in the ORD pilot [36] on the understanding that Open Data is a mandatory requirement, whereas the actual requirement is "as open as possible, as closed as necessary". Such misunderstandings have serious implications given that recognition in the sciences relies largely on knowledge discovery, which in turn relies on data. It is thus unsurprising that a researcher who has invested significant effort in designing and undertaking data creation activities may be reluctant to make the results openly accessible and thereby risk a loss of intellectual advantage. Similarly, for commercial organisations the loss of advantage associated with Open Data risks losing competitive advantage. Hence, market competition not only discourages partners from making their data open but also risks companies not participating in the ORD pilot. While data accessibility should not present any particular difficulties because of the "as open as possible, as closed as necessary" principle, in reality academic and commercial sensitivities are such that any suggestion of Open Data being mandatory will prove an obstacle to data management. This is somewhat unfortunate given that restricting access to data to mitigate loss of competitive advantage is entirely compatible with the FAIR principles [18].

Beyond the authoring of the DMP, ill-considered programming of tasks also has the potential to detract from the data management activities. Whereas there is a tendency for data management tasks to be a feature of dissemination and exploitation work packages, any circumstance that decouples data collection from data creation will likely hinder the data management activities of a project. Instead, data collection tasks need to be a feature of the activity that generates the data, otherwise there will be the risk of delays and hence a failure to share data in a timely manner amongst project partners.

*3.2. Solutions*

Data management has to overcome various challenges if its potential benefits are to be realised. While underlying issues are invariably inter-related, the authors give attention to the issues individually.

*Research culture*—there are good reasons to believe that research data management will (or perhaps already has) become a feature of mainstream research insofar as the entities upstream and downstream of research, namely the funding agencies and the publishing houses, respectively are mandating improved data management practices. What remains is for the research community to be convinced that data management adds value to the research process. In anticipation of realising such a circumstance, perhaps most important is to demonstrate that, rather than simply delivering a body of data towards the project term, data management supports the scientific objectives of a project. Simply delivering a body of data does not contribute to the scientific objectives of a project and can reasonably be claimed to consume resources better used for research. Instead, where data management is an integral feature of data creation activities from the outset of a project, opportunities will exist for exchanging data between partners as they become available; ensuring data consistency and quality; and for entering into data sharing arrangements. With systematic research data management being a relatively new phenomenon, concrete examples of the benefits are sparse. That said, over the relatively short period of the H2020 ORD pilot, the authors have direct experience of both anticipated and unforeseen benefits. For example, in consequence of data peer review, the INCEFA-PLUS project was able to deliver a body of high-quality test results that provided the basis for data mining activities and an international data sharing arrangement in a subsequent H2020 proposal, which has since been favourably evaluated [37]. Presently however, it is still not uncommon to encounter a combination of a lack of data sharing culture and motivation [17,24]. Sometimes in fact, the authors have encountered even quite hostile reactions and in such circumstances the failure to adhere to the DMP becomes a self-fulfilling prophecy.

Given the DMP is a relatively recent phenomenon that is still somewhat unfamiliar to the research community, a gradual introduction to its concepts and purpose is required that will allow researchers to become aware of potential pitfalls and benefits. This circumstance favours engagement with early-career researchers, who by definition are in the process of learning the skills needed for a career in the sciences. Early-career researchers are also likely to be more favourably inclined due to their not having been exposed to a research environment where data management has been entirely lacking. Hence, giving responsibility for the DMP and data management tasks to early-career researchers would provide a concrete opportunity to contribute both to their career development and to the implementation of the projects in which they are participating.

For consortia as a whole, it is important that all partners participate in the DMP activities. Just one weak link and the data management risks being undermined. In this context, one value proposition that may motivate participation is the development of organisational data documentation standards and protocols, which have the potential to improve data quality, facilitate cross-department data exchange and minimise data loss within an organisation. In particular, thorough documentation of the data lifecycle at any research phase provides all the details about their generation, management and processing, thereby allowing validation of the research results. This and other potential

benefits, such as data sharing and data citation, could be the subject of training workshops that feature as milestones of any given project.

In the context of EC-funded projects where the JRC has participated in a data management role, the appointment of a Data Committee has proven particularly effective in engaging researchers in data management activities [38,39]. Perhaps most importantly, the collective responsibility for authoring and maintaining the DMP ensures that the larger proportion of the project partners are familiar with and supportive of the data policy of the project. Furthermore, by undertaking actions such as the formulation of data protocols and data review, partners become familiar with the practicalities of DMP implementation and the accompanying added value, whereby data protocols identify the data needed to achieve scientific objectives and data review ensures the quality of data and helps ensure the consistency of the data coming from the different partners.

*Data sharing*—following the success of the FP7 Open Access pilot, it is unsurprising that funding agencies have sought a similar access paradigm for research data. Certainly, data can be completely open and freely available for reuse, such as public sector data, which although typically lacking inherent value, can yield derived value, such as services developed to locate preferred parking zones based on municipal parking fine data or suitable bathing areas based on municipal water quality data [40]. There will though be legitimate reasons for data remaining entirely closed e.g. where there is inherent commercial, financial or intellectual value. To address such circumstances, i.e. where consortia consider that little or no data are available or suitable for open access, the European Commission allows complete withdrawal from the ORD pilot (commonly known as opt-out) at any stage during the project lifecycle, even after the Grant Agreement has been signed. While the authors have experience both of both circumstances, typically data will be positioned somewhere between the extremes of the data access spectrum. Again, the European Commission recognises this circumstance and although promoting open access to research data is a core principle of the H2020 ORD pilot, making the entirety of project data open has never been mandatory [41]. Specifically, the principle of "as open as possible, as closed as necessary" allows for keeping certain data sets open and others closed. Even so, the notion that data access can simply be open or closed ignores the complexities of data sharing. Hence, further measures may be required.

Data may be open but the owner expects acknowledgement in derivative works, which would require that the data are accompanied by bibliographic data and a licence. Conversely, data may be nominally closed but made available if sharing is on mutually beneficial terms, which would require the data can be discovered and access requests submitted. Both these circumstances can be addressed by enabling data for citation. Irrespective of whether open or closed, data citation ensures the data can be discovered and the data creators acknowledged. Beyond these benefits, citing data also allows for data transparency, thereby facilitating the verification of results. Furthermore, where data citation relies on a digital identifier, such as the digital object identifier (DOI), there is scope for machine-readability and long-term preservation. Identifiers ensure the data with which they are associated can be discovered irrespective of usage licences. The practice of citing data typically relies on infrastructures that generate and assign DOIs and store metadata for data sets. Data repositories that register identifiers automatically via such infrastructures are becoming more common, e.g. Zenodo [33] assigns DataCite DOIs for all uploads [22]. Similarly, in the context of those EC-funded projects where the JRC has participated, extensive use has been made of the DataCite framework. The DataCite metadata schema ensures sufficient bibliographic data are available for citation and that the terms of (re)use are specified. The platforms that DataCite operates expose the metadata in various formats, thereby promoting discoverability and reuse [42,43].

Using the DataCite framework, the JRC engineering materials database hosted at https://odin.jrc.ec.europa.eu (accessed 30 October 2020) allows data to be cited in the same way as traditional scientific publications, so that Open Data remain open and closed data remain closed, but all data sets are citable and discoverable by way of their publicly

accessible bibliographic metadata. In the case of nominally closed data, the platform supports submitting data access requests. This allows bilateral discussion and informed decision by both parties about whether to enter into data sharing. Consequently, all circumstances where data owners may have reservations about data sharing can be accommodated, whether it is because of the complexities of the data, the protocols used to produce the data; the commercial sensitivity of the data; etc. Where there is an underlying willingness to share data, this combination of data citation and on-demand data access delivers a "managed" data access paradigm that can reasonably be claimed to be more effective than that of Open Data, accommodating as it does the reuse of both open and closed data; the competitive interests (intellectual and commercial) of project partners; opportunities for dialog between the owners and consumers of the shared data. In those projects where the JRC has participated, "managed" data access has alleviated the data sharing concerns of industrial partners, thereby encouraging their participation in the data management activities, typically more so than partners from the academic sector, where it appears that concerns about the increased transparency are an obstacle to data sharing.

Similarly, in those EC-funded projects where IRES has participated, effective data management practices ensure data generated during a project are gathered and stored for use over the entire term of the project and reuse thereafter. In this context, ensuring that the data are citable will promote their discovery; the transparency of project results; reuse by other members of the research community; and accreditation [44]. The H2020 eNanoMapper project is a case that strongly supports this argument. eNanoMapper developed a computational infrastructure for the management of engineered nanomaterials toxicological data by integrating related public data stored in several databases [45].

Enabling access to scientific data may also add credibility to the results of a project, whereby data can be validated by scientists from outside organisations and institutes. In consequence, the reputation of organisations that deliver quality data increases. Additionally, the existence of large and well-organised bodies of data that result from effective data management may also become a useful training set for algorithms, which can be exploited to the benefit of the owning organisation(s) as well as third parties.

The experience of the authors has been that academic and industrial partners can be unwilling to share their data publicly in the circumstance their competitive position in their respective fields suffers. In this circumstance, the benefits that accompany open access are insufficient to counter lost advantage. In which case, alternative data sharing paradigms need to be explored, such as restricted data sharing or data licensing, which may constitute a significant value proposition for the data owner. Restricted data sharing refers to the exchange of data between the members of a specific group or project consortium to their mutual benefit. Otherwise, data licensing can be adopted to promote data brokerage, whereby a transaction takes place, either in-kind by reciprocal data exchange or financial. The data licensing approaches have the potential to become increasingly popular, especially due to the development of data marketplaces, such as those of the VIMMP [46] and MarketPlace [47] projects. Again however, data integration with marketplaces requires standardised vocabularies and technical infrastructure that can ensure data interoperability with minimal effort for the data provider. As long as these requirements remain unresolved, these approaches cannot easily be pursued.

*Infrastructure*—open data standards enable interoperability and the development of a robust infrastructure that supports data management activities, whereby interoperability is feasible between humans, humans and machines and just machines. Such communication will face critical challenges in the circumstance that incompatibilities exist in the language used to express the meaning of the data and in the manner that data are collected, stored and shared.

For data collection, measures have to be taken that will determine the personal data to be shared and recommendations for predetermined procedures and methods to be adopted for the acquisition and handling of the research data for the duration of a project. Additionally, people and organisations that exchange data will form heterogeneous

network systems i.e. networks that consist of computers and other devices that use different protocols and configurations. Moreover, the data that is exchanged may also be heterogeneous. To address these circumstances, network architectures and communication technologies are needed that enable the interconnectivity of the devices and the seamless transfer of digital data. Such specifications may concern the file format, the data structure and the methods used for data exchange, including the ways in which access is granted.

Efficient data exchange requires standards for the terminology used to describe data and metadata. Such vocabulary harmonisation is important since misunderstandings about the meaning of words can lead to incorrect assumptions. A shared vocabulary that consists of words that are rigorously defined helps to communicate knowledge accurately. Aligned to the development of vocabularies is the development of ontologies, which are formal descriptions of knowledge that consist of concepts and relationships. It is not only important that new domain ontologies be developed but also that existing ones are maintained and expanded. Where different terms and relationships are used to represent data, means exist to create connections among them, such as ontology alignment methods, which map entities from one ontology to another. Furthermore, shared vocabularies along with ontologies need to be machine-readable, thereby rendering the sharing of data among machines easier and allowing for further computational analysis. Hence, semantic interoperability does not only contribute to a common understanding of the knowledge among the partners but also to the enhanced inference of information by machines.

A DMP can begin to address interoperability insofar as data management activities should employ already existing standards. Thereafter, DMP tasks can extend to the development of new standards for those domains and applications where data harmonisation is limited or non-existent. A case where the DMP included such initiatives is that of the OYSTER H2020 project [28], where a standardised vocabulary was developed for the description of materials characterisation methodologies and the resulting data. OYSTER also undertook to harmonise the process of the DMP, by creating the Data Management Ontology [34]. This ontological representation, where the entities of data management and materials characterisation co-exist in a unified linked data schema, can serve as a paradigm for harmonisation of the high level DMP concepts with the domain specific terminology. Such data representations from various domains can be used to standardise the data management process even further. In this regard, the Data Committee responsible for the DMP would operate a software product built on those data structures to allow for the integration of the project data with either open access repositories or data marketplaces. Moreover, the Data Committee consults the partners on the operation of the system and supervises the data integration process. This approach would accommodate both public data sharing and data licensing/brokerage, as well as allow the DMP to deliver more predictable and effective results. Currently, although such a digital infrastructure does not exist, its development could be considered important given the EC investment in data marketplaces, open innovation environments and open innovation test beds. It would be reasonable to envision such a software product as an extension of the CORDIS platform, with its development funded by one or more calls of Horizon Europe (HE).

In those EC-funded projects where the JRC participated in a data management role, consortia have been encouraged to develop interoperability standards for data, whereby solutions are implemented for test types of immediate interest in accordance with a common methodology. In so doing, as well as addressing its data exchange and systems integration requirements, each project contributes to a body of data standards of use to the wider engineering materials community. Such work has typically taken place in the context of the standards setting environment, with CEN Workshops providing a platform to engage a broad representation of stakeholders, promote widespread adoption and address the longer-term prospects of the resulting technologies. While these standards serve the needs of any given project, their use by project partners serves to validate the

standards and allow their further development to an increased technology readiness level. The result has been the delivery of standards for fatigue and nanoindentation test data, where the technical specifications are described in CEN Workshop Agreements [48,49] and corresponding reference implementations are publicly available from http://uri.cen.eu/cen (accessed 30 October 2020). For example, the reference implementation for fatigue test data is available as an XML Schema Definition from http://uri.cen.eu/cen/cwa/17157/1/xsd/iso-12106.xsd (accessed 30 October 2020).

Beyond addressing the technical challenges of developing interoperability standards for data, The CEN Workshops have given particular attention to the implications for the industrial and standards setting communities, whereby data standards have the potential to be disruptive. In the standards setting domain for example, the workflows and publication routes for ICT standards differ to those of traditional, documentary standards, with the result that entirely new policies are presently the subject of attention at both CEN and ISO. In those EC-funded projects where IRES has participated, CEN Workshops have also provided the platform for the development of documentation for material modelling data [50] and material characterisation data [51]. This trend continues with a number of recently funded H2020 projects in which the authors are participating, including nanoMECommons, OntoCommons and OpenModel, whereby interoperability standards and domain ontologies will be delivered in support of data exchange, semantic interoperability and data science as such, now adopting materials modelling and characterisation data workflows templates [52–55].

*Project co-ordination*—overcoming practical issues that arise during proposal preparation and project implementation will mitigate the risk associated with the data management policies of any given project. As indicated already, potential difficulties extend to misunderstandings about data management roles and responsibilities, adoption of practices tailored to project needs and the allocation of the necessary resources.

There is a need for a precise description of the data management roles and responsibilities of everyone involved. Responsibilities may concern the collection and curation of the data, their upload to the platform used for their storage, the reporting of research progress and the backup of the data in case of incidents that endanger their security. The clear description of the role of each partner in the DMP, as well as the corresponding responsibilities, should be specified in the proposal. In practice, this can be managed by anticipating the establishment of a Data Committee, so that during proposal preparation, individual partners have an opportunity to enrol in the Data Committee and hence give consideration to their contributions. While it would be neither effective nor appropriate for all partners to participate, the membership of the Data Committee should be sufficiently representative to ensure the consortium as a whole becomes aware of their roles and responsibilities. Moreover, the partner responsible for data management should communicate this information to the other partners as early as possible during proposal preparation. In this regard, partners have to be aware of the data sets to which they have access. Furthermore, consideration is needed of the legal restrictions concerning the dissemination and use of the data after the end of a project. Such policies should define the data to be shared; the duration of their availability; whether they will be open, closed or restricted; and their use by third parties. These policies ensure that research data will be used ethically and with due respect to any personal or sensitive information. Establishing a common understanding about the accessibility of research data is also important.

To encourage participation in the DMP, the effort required from the project partners needs to be minimised. To gather the information required for the synthesis of an effective DMP, one option is to undertake a survey of the partners. It is important that any such survey consist of questions tailored to the data management needs of the specific project. Simple questionnaires targeted towards the needs of a project in combination with comprehensive guidance not only prevents the researchers getting distracted from the execution of their tasks but also renders data collection easier. Thereafter, the template provided by the EC for H2020

projects is sufficiently general to be applicable to any type of research project but can be adjusted to serve a particular case based on the results of the survey.

To allow for the implementation of all DMP tasks, explicit estimations of resourcing should be provided at the time of the proposal. Awareness of the required budget prior to the start of a project helps organisations manage their resources effectively. Such practice also allows identification of mismatches between project tasks regarding the allocation of resources [56] and may prevent misunderstandings among the partners. Examples of the costs that have to be taken into consideration for data management include the personnel time for data entry tasks and participation in the Data Committee; platform design and development; accompanying hardware; and archival beyond the term of the project since there may be charged fees depending on the storage duration.

## 4. Conclusions

The ORD pilot clearly resonates with the European scientific community. With the initial implementation targeting a few select calls but allowing for voluntary participation, as of 2017 the pilot was rolled out across the entire research programme. Although the extent of participation in the ORD pilot provides a concrete indicator of its success, the experience of the authors suggests scope for improvement, particularly at the proposal stage. Whereas commitments to collect and share project data tend to be favourably received during proposal evaluation, this needs to be accompanied by a critical review of the roles, responsibilities and resourcing estimates required to achieve those commitments. Further, for improved research data management practices to be realised, circumstances under which the DMP is the responsibility of a sole partner should be avoided. Instead, a broad representation of project partners needs to be engaged in the formulation and implementation of the DMP, preferably by way of participation in a Data Committee with well-defined roles and responsibilities. Further, explicit tagging of data management tasks and deliverables and examination of the corresponding resourcing will provide clear insights into the credibility of the data management component of the project.

Without exception, the authors advocate data management in the sciences. Their experience from having led the data management strands of various H2020 projects argues in favour of a pragmatic approach that aims at improved data management practices over adherence to any specific data access model. While the interpretation of the experiences is subjective and certainly open to debate, indications are that prioritisation of Open Data in the sciences risks being counterproductive and that attention should instead focus on data management and the needed infrastructures becoming an established feature of the mainstream research process. Furthermore, realism is required both in respect of the funding of data management tasks and the expected impacts. In the first instance, tasks delivering the required infrastructure need to be prioritised and receive the requisite funding. In this context and where appropriate, the DMP should extend to the development of interoperability standards, whereby a common data documentation approach adopted by any specific project is disseminated to a broader audience, including standardisation bodies.

The success of the Open Access initiative, which can reasonably be claimed to have motivated improved research data management practices, may have resulted in too much emphasis on the data access model at the detriment of establishing a data management ecosystem capable of supporting the research process and driving knowledge discovery and innovation. Although the H2020 guide emphasises that data should be "as open as possible, as closed as necessary", experience has been that project partners interpret the title of the pilot as mandating Open Data. Thereafter, such misunderstandings risk some partners being reluctant to participate in the data management activities, with the result that the entire data management component of a project is undermined. Instead, where it has been understood that tailoring the data access model to the needs of the project, its research objectives and the intellectual and/or commercial interests of the project partners

is entirely permissible in the context of the FAIR data principles, data management has flourished and project partners (and hence all stakeholders) have benefitted.

Demonstrable examples of the added value of the ORD pilot include closer co-operation facilitated by data sharing, improved data quality and opportunities for new projects made feasible by the existence of a large body of research data. Where issues have been encountered, these can generally be attributed to insufficient resourcing, misunderstandings about roles and responsibilities, failure to collect and share data from the outset of the project and concerns about the implications of data sharing both within and beyond the project consortium. With many participants in multi-partner research projects often lacking data management experience, the main future prospective of the reported work is to increase partner (both academic and industrial) awareness and participation in the research data management process and ultimately increase the potential for data sharing in subsequent research framework programmes.

To conclude, in those projects where data management has been embraced, there have been concrete beneficial impacts. Where problems have been encountered, these can largely be attributed to cultural and technical obstacles and there is good reason to expect these can be entirely alleviated. Whereas there is scope for extensive recommendations [12], the authors tend towards just five, as follows:

- Give due consideration at the proposal preparation stage to roles, responsibilities and resourcing and to establishing a Data Committee to take responsibility for leading data management activities;
- Assign data management responsibilities to early-career researchers, thereby enhancing career development and promoting the longer-term prospects of research data management;
- Employ data citation to promote data discovery, accessibility and sharing;
- While giving due attention to the H2020 principle that data should be "as open as possible, as closed as necessary", tailor the data access model to the needs of the project, its research objectives and the competitive interests (intellectual and commercial) of project partners;
- Engage in the development and/or utilisation of interoperability standards that are tailored to the immediate interests of the project but also complementary to community initiatives.

**Author Contributions:** Investigation, T.A., K.B., T.E. and E.P.K.; resources, E.P.K.; writing—original draft preparation, T.A., K.B., T.E. and E.P.K.; writing—review and editing, T.A., K.B., T.E. and E.P.K.; visualization, K.B., T.E. and E.P.K.; supervision, E.P.K. All authors have read and agreed to the published version of the manuscript.

**Funding:** The paper partially received funding from the European Union under Grant Agreement numbers 760827 (OYSTER), 760779 (SMARTFAN), 814505 (DECOAT), 814588 (REPAIR3D) and 814552 (LIGHTME). Those projects where the JRC has participated have received funding from the Euratom research and training programme 2014–2018 under Grant Agreement numbers 662320 (INCEFA-PLUS), 755039 (M4F), 755151 (MEACTOS), 755269 (GEMMA) and 945300 (INCEFA-SCALE).

**Institutional Review Board Statement:** Not applicable.

**Informed Consent Statement:** Not applicable.

**Data Availability Statement:** Data available in a publicly accessible repository (temporarily offline at the time of publication due to maintenance).

**Conflicts of Interest:** Tim Austin is an employee of the European Commission. Otherwise, the authors declare no conflict of interest.

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
