# Peer review of "Lessons Learnt from Engineering Science Projects Participating in the Horizon 2020 Open Research Data Pilot"

_data, 2020_

Round 1
Reviewer 1 Report
This paper examines the data management activities and outcomes of a number of projects participating in the Horizon 2020 Open Research Data pilot. The result has been to identify a number of commonly-encountered benefits and issues; to assess the utilisation of data management plans; and by close examination of specific cases to gain insights into obstacles to data management and potential solutions. Although primarily anecdotal and difficult to quantify, the experiences reported in this paper tend to favour developing data management best practices rather than doggedly pursue the Open Data mantra. While Open Data may prove valuable in certain circumstances, there is good reason to claim that managed access to scientific data of high inherent intellectual and financial value will prove more effective in driving knowledge discovery and innovation.
This is a well written manuscript with a thorough reference list that is much desired in the community. However, the manuscript leaves a lot to be desired.
Comments/Questions:
1. Can the authors point a quantitative measure of "FAIRness" for every publication? Can the authors recommend other quantitative ways (analogous to various defined impact factors) to identify the level of fairness in every publication or/and project?
2. The authors mention various engineering databases, for example Refs.44,45 for fatigue and nanoindentation. The manuscript contains many other libraries' mentions, an example of which is the JRC one. However, in my example effort to access the database of Ref.45, it was impossible to figure out how to read the reported data. Can the authors summarize how to access *every* link and what to do, how to navigate it, when access it, so that it becomes clear what the usefulness is?
3. Can the authors provide some example cases from projects (without uncovering any personal data) so that particular applications become very clear to the reader. Some flow diagram figures would be very helpful. For example, the authors mention DataCite uses, but the information provided is not self-contained in a way that the reader becomes aware of benefits, uses, disadvantages in various characteristic examples.
4. A paper about data cannot just include a table filled with text. Figures and example datasets from exemplary cases that point out what FAIR means, and how it should be used, are mandatory. The authors are invited to supply the manuscript with at least 2 example cases, including flow diagrams and datasets examples for each of the cases.
5. English should be paid attention to. For example: "mature data management best practices" is grammatically messy.
Author Response
The authors are grateful for the comments and have endeavoured to implement revisions accordingly. Replies to the individual comments, as follows:
R1C1. Can the authors point a quantitative measure of "FAIRness" for every publication? Can the authors recommend other quantitative ways (analogous to various defined impact factors) to identify the level of fairness in every publication or/and project?
A new reference [18] about FAIRness metric development is introduced. This aside, FAIR principles are by definition specific to data, so it is assumed the reference by the reviewer to ‘publication’ is intended to mean data publication. Previously, the FAIR Maturity Evaluation Service at https://fairsharing.github.io/FAIR-Evaluator-FrontEnd has yielded a score of about 50% for data hosted at the JRC ODIN Portal. The comment is answered by making reference in the manuscript to the results of two such evaluations.
R1C2. The authors mention various engineering databases, for example Refs.44,45 for fatigue and nanoindentation. The manuscript contains many other libraries' mentions, an example of which is the JRC one. However, in my example effort to access the database of Ref.45, it was impossible to figure out how to read the reported data. Can the authors summarize how to access *every* link and what to do, how to navigate it, when access it, so that it becomes clear what the usefulness is?
As per the sentence from the manuscript reading ‘The result has been the delivery of interoperability standards for fatigue and nanoindentation test data, the specifications for which are publicly accessible from http://uri.cen.eu/cen and described in CEN Workshop Agreements [44,45].’, references 44 and 45 are to technical specifications for data standards, not databases. The references describe the data specifications in detail. The link in the text is to the platform that hosts various data specifications, including those corresponding to references 44 and 45. The comment has been addressed by adding some further explanation, including a link to the data specification for reference 44, e.g. http://uri.cen.eu/cen/cwa/17157/1/xsd/iso-12106.xsd.
R1C3. Can the authors provide some example cases from projects (without uncovering any personal data) so that particular applications become very clear to the reader. Some flow diagram figures would be very helpful. For example, the authors mention DataCite uses, but the information provided is not self-contained in a way that the reader becomes aware of benefits, uses, disadvantages in various characteristic examples.
The authors consider the benefits and uses of DataCite are described by the text reading ‘In the context of creating FAIR data, experience suggests that data citation is a particularly effective means for promoting improved data management practices, whereby any data set, whether restricted or open can be discovered. Thereafter, reuse of data can be properly managed (by way of licensing) and accredited (by way of entries in the references section of a traditional report or scientific publication). Data citation thus entirely addresses the findable and accessible strands of the FAIR principles and motivates data sharing (and hence reuse) by ensuring accreditation.’
A flow diagram for the workflow of the Repair3D/Decoat project is added. Other examples exist that could be presented in flow diagram form but the relatively short revision cycle has left insufficient time. For example, there are INCEFA-PLUS publications referenced in the manuscript that touch on the data peer review workflow, whereby data for different classes of information are uploaded to the project database; linked to the primary test result; reviewed by a subject matter expert at the organisation that undertook the test; revised as necessary and validated i.e. published; reviewed by an expert panel (aka Data Committee) consisting various partners from the project consortium; enabled for citation in the circumstance required quality and completeness criteria are satisfied, otherwise unvalidated i.e. unpublished in anticipation of the creator organisation making revisions are the recommendation of the expert panel prior to another review. A second review cycle would allow the authors the opportunity to elaborate the workflow and introduce a corresponding flow diagram.
R1C4. A paper about data cannot just include a table filled with text. Figures and example datasets from exemplary cases that point out what FAIR means, and how it should be used, are mandatory. The authors are invited to supply the manuscript with at least 2 example cases, including flow diagrams and datasets examples for each of the cases.
This comment is addressed by revisions made in response to the 1st and 3rd comments.
R1C5. English should be paid attention to. For example: "mature data management best practices" is grammatically messy.
The intended meaning for a native English speaker of ‘mature data management best practices’ is best practices in the domain of data management that are maturing or have matured. With this in mind, the text of the manuscript seems reasonably clear and succinct. Possibly the issue the example aims to highlight is redundancy insofar as best practices may be considered to be mature by definition, in which case either ‘mature’ or ‘best’ can be removed. Although the issue the example highlights is not entirely understood, it has been addressed in the context of an editorial review of the entire paper. If the comment is not addressed to the satisfaction of the reviewer, the authors are perfectly willing to give further attention to the comment in the circumstance the reviewer is able to elaborate their concerns.
Reviewer 2 Report
Journal: Data (ISSN 2306-5729)
Manuscript ID: data-1319410
Title: Lessons Learnt from Engineering Science Projects Participating in the Horizon 2020 Open Research Data Pilot
In this paper, the authors tried to figure out the data management as a feature of the mainstream research process. The result has been to identify a number of commonly-encountered benefits and issues; to assess the utilisation of data management plans; and by close examination of specific cases to gain insights into obstacles to data management and potential solutions. Although primarily anecdotal and difficult to quantify, the experiences reported in this paper tend to favour developing data management best practices rather than doggedly pursue the Open Data mantra. While Open Data may prove valuable in certain circumstances, there is good reason to claim that managed access to scientific data of high inherent intellectual and financial value will prove more effective in driving knowledge discovery and innovation.
The research is of great interest. The following manuscript has some weakness and here are the most important topics/questions to be dealt with:
1. Please rewrite the abstract by identifying the purpose, the problem, the methodology and the important results (not all) and conclusions of your work.
2. Introduction
The Introduction should consist of five paragraphs answering the following five questions:
What is the problem?
Why is it interesting and important?
Why is it hard?
Why hasn't it been solved before? (Or, what's wrong with previously proposed solutions?)
What are the key components of my approach and results?
As the above-mentioned questions should be replied to, it will be better that write more relation to the benefits & disadvantages of the blending techniques and more investigate about limitation of previous studies. Also, this part needs more explanations to state clearly the objectives & hypothesis of this study at the end of the Introduction part. It should be mentioned to the factors that be shed light by this study.
3. Is there a possibility to show some figures in the manuscript based on the authors' studies for EU projects?
4. Conclusion section is extremely long. The conclusions are very weak and ít requires a deeper analysis of the results.
The reviewer suggests carefully read the whole manuscript again before resubmitting it to the journal Data. Authors should consider the above-mentioned remarks in order to revise the manuscript. The reviewer thinks that a publication of the draft manuscript may be possible after a minor revision.
Author Response
The authors are grateful for the comments and have endeavoured to implement revisions accordingly. Replies to the individual comments, as follows:
R2C1. Please rewrite the abstract by identifying the purpose, the problem, the methodology and the important results (not all) and conclusions of your work.
The abstract has been drafted in accordance with the guidelines of a training course given by a highly experienced STM publishing executive with over 35 years experience (https://melitapublishingconsultants.com). With this in mind, the authors kindly ask to know the shortcomings of the present structure before implementing a revised structure.
R2C2. Introduction
The Introduction should consist of five paragraphs answering the following five questions:
What is the problem?
Why is it interesting and important?
Why is it hard?
Why hasn't it been solved before? (Or, what's wrong with previously proposed solutions?)
What are the key components of my approach and results?
As the above-mentioned questions should be replied to, it will be better that write more relation to the benefits & disadvantages of the blending techniques and more investigate about limitation of previous studies. Also, this part needs more explanations to state clearly the objectives & hypothesis of this study at the end of the Introduction part. It should be mentioned to the factors that be shed light by this study.
The introduction conforms to a structure similar to that of the abstract. It has though been updated to take into consideration the comments of the reviewer.
R2C3. Is there a possibility to show some figures in the manuscript based on the authors' studies for EU projects?
Two figures are added, one of which is a flow diagram for the workflow of the Repair3D/Decoat project is added. Other examples exist that could be presented in flow diagram form but the relatively short revision cycle has left insufficient time. For example, there are INCEFA-PLUS publications referenced in the manuscript that touch on the data peer review workflow, whereby data for different classes of information are uploaded to the project database; linked to the primary test result; reviewed by a subject matter expert at the organisation that undertook the test; revised as necessary and validated i.e. published; reviewed by an expert panel (aka Data Committee) consisting various partners from the project consortium; enabled for citation in the circumstance required quality and completeness criteria are satisfied, otherwise unvalidated i.e. unpublished in anticipation of the creator organisation making revisions are the recommendation of the expert panel prior to another review. A second review cycle would allow the authors the opportunity to elaborate the workflow and introduce a corresponding flow diagram.
R2C4. Conclusion section is extremely long. The conclusions are very weak and ít requires a deeper analysis of the results.
The reviewer suggests carefully read the whole manuscript again before resubmitting it to the journal Data. Authors should consider the above-mentioned remarks in order to revise the manuscript. The reviewer thinks that a publication of the draft manuscript may be possible after a minor revision.
An editorial review of the entire manuscript has included a revision of the conclusions.
Reviewer 3 Report
The manuscript presents very good work related to Engineering Science Projects and going to be interesting for the readers.
Some minor comments are as follows.
- Authors need to include some interesting data in the abstract part of the manuscript.
- English must be improved.
- Novelty of the work be established.
- All the important results reported be compared in a tabular form to establish the superiority of the work.
- Authors need to add future prospective of presented research in the conclusion part of the manuscript.
- Authors must need to incorporate recent reference related to subject of the manuscript to make it more interesting for the readers.
Author Response
The authors are grateful for the comments and have endeavoured to implement revisions accordingly. Replies to the individual comments, as follows:
R3C1. Authors need to include some interesting data in the abstract part of the manuscript.
As an alternative to including interesting data in the abstract, which is already quite lengthy, the authors have made reference to quantitative FAIR evaluations in the body of the manuscript.
R3C2. English must be improved.
An editorial review of the entire manuscript has given attention to the language.
R3C3. Novelty of the work be established.
The experience of the authors when participating in EU-funded projects is that partners have very varied perspectives and approaches to data management. This is suggestive of a lack of scientific literature that documents the outcomes of data management initiatives. Hence the absence of best practices, which is the shortcoming the work attempts to address. To the best of our knowledge, no relevant work on experience from DMP methodologies in EU projects (not reporting DMP as such) is reported in literature.
R3C4. All the important results reported be compared in a tabular form to establish the superiority of the work.
Given that the manuscript does not report research outputs as such, the authors are not entirely sure how to address this comment. If the reviewer is able to elaborate, the authors can give attention to the comment in a subsequent revision cycle.
R3C5. Authors need to add future prospective of presented research in the conclusion part of the manuscript.
The conclusions have been updated to explain that with participants in multi-partner research projects often lacking data management experience, the main future prospective of our work is to increase partner (both academic and industrial) awareness and participation in the research data management process and ultimately increase the potential for data sharing in subsequent research framework programmes.
R3C6. Authors must need to incorporate recent reference related to subject of the manuscript to make it more interesting for the readers.
To respond meaningfully to this comment, more time would be needed to undertake an exhaustive literature review. For the moment, we have been unable to find other similar papers, which to some extent supports the reply to the comment on needing to establish the novelty of the work.
Round 2
Reviewer 1 Report
I believe the authors addressed adequately the suggestions made.